# Novel Insights in Venous Thromboembolism Risk Assessment Methods in Ambulatory Cancer Patients: From the Guidelines to Clinical Practice

**DOI:** 10.3390/cancers16020458

**Published:** 2024-01-21

**Authors:** Anca Drăgan, Adrian Ştefan Drăgan

**Affiliations:** 1Department of Cardiovascular Anaesthesiology and Intensive Care, Emergency Institute for Cardiovascular Diseases “Prof. Dr. C C Iliescu”, 258 Fundeni Road, 022328 Bucharest, Romania; 2Faculty of General Medicine, Carol Davila University of Medicine and Pharmacy, 8 Eroii Sanitari Blvd, 050474 Bucharest, Romania; dragan.adrian.stefan24@gmail.com

**Keywords:** venous thromboembolism risk, ambulatory cancer, risk assessment

## Abstract

**Simple Summary:**

Cancer patients are at greater risk of developing venous thromboembolism compared to the general population, which can lead to a decreased quality of life, a worsened prognosis, and increased treatment costs. Guidelines provide clear strategies for preventing thrombosis in hospitalized cancer patients and those undergoing surgery. For ambulatory cancer patients, thromboprophylaxis is recommended only for those who are at high risk. However, this can be challenging in clinical practice. The current guidelines do not provide sufficient information on this problem. Imaging and biomarker screening techniques are underutilized in practice. Although new risk scores, nomograms, and strategies have been developed using biomarkers and clinical and genetic features, many of these methods have not yet been validated. Machine learning algorithms have already been studied with promising results. This review presents the current knowledge on venous thromboembolism risk assessment in ambulatory cancer patient settings.

**Abstract:**

Many cancer patients will experience venous thromboembolism (VTE) at some stage, with the highest rate in the initial period following diagnosis. Novel cancer therapies may further enhance the risk. VTE in a cancer setting is associated with poor prognostic, a decreased quality of life, and high healthcare costs. If thromboprophylaxis in hospitalized cancer patients and perioperative settings is widely accepted in clinical practice and supported by the guidelines, it is not the same situation in ambulatory cancer patient settings. The guidelines do not recommend primary thromboprophylaxis, except in high-risk cases. However, nowadays, risk stratification is still challenging, although many tools have been developed. The Khrorana score remains the most used method, but it has many limits. This narrative review aims to present the current relevant knowledge of VTE risk assessment in ambulatory cancer patients, starting from the guideline recommendations and continuing with the specific risk assessment methods and machine learning models approaches. Biomarkers, genetic, and clinical features were tested alone or in groups. Old and new models used in VTE risk assessment are exposed, underlining their clinical utility. Imaging and biomolecular approaches to VTE screening of outpatients with cancer are also presented, which could help clinical decisions.

## 1. Introduction

Cancer patients often present with a prothrombotic state due to the abnormalities in each component of Virchow’s triad, thus contributing to thrombosis. Researchers estimated that VTE would occur in 4–20% of cancer patients at some stage, with the highest risk immediately following cancer diagnosis [1]. In the last period, the VTE incidence in oncologic patients has increased in the context of the higher performance of imaging techniques and the development of new cancer treatments that improved survival [2]. After cancer diagnosis, the 12-month cumulative VTE incidence was 3%, a percentage nine times higher when compared to the general population [2]. 

However, despite improved cancer treatment, VTE in cancer patients is strongly associated with a poor prognosis. The cumulative mortality in VTE cancer patients was 27.7% after one month, 48.7% after three months, 68.2% at one year, and 84.1% after five years, which is much higher than the cumulative mortality in cancer patients without VTE (7.5%, 17%, 38.5%, and 84.1%, respectively) [3]. Pulmonary embolism (PE) was associated with a poorer prognosis than venous thrombosis [3]. The one-year mortality of the PE cancer patients was 73% in Sørensen et al.’s study, as compared to 39.3% in the non-cancer cohort [3]. 

Khorana et al. reported that 17.1% of the patients recently diagnosed with cancer and with VTE events would develop recurrent episodes of VTE during a nine-month follow-up period [4]. The total costs related to the healthcare of the patients with VTE recurrence were very high, suggesting the necessity of reducing VTE risk in cancer patients [4].

Thromboprophylaxis in hospitalized cancer patients and perioperative settings is widely accepted in clinical practice and supported by the guidelines. However, most cancer patients would develop VTE in the outpatient setting. Primary thromboprophylaxis is not routinely recommended, except for high-risk cancer patients. Selecting an ambulatory cancer patient who would benefit from thromboprophylaxis is still challenging because of the specific bleeding risk. The Khrorana score is mainly the recommended tool in this setting, but many limits of this old score have been reported. Novel approaches have been proposed. Clinical features, routine hematologic and coagulation lab testing, new biomarkers, and genetic data, separately or grouped, were introduced in the novel risk scores, nomograms, or machine learning algorithms to accurately assess the VTE risk in ambulatory cancer patients in general and specific tumors. This narrative review aims to present the current relevant knowledge in this setting starting from the guideline recommendations and continuing with the specific risk assessment methods to help clinicians in their decision regarding primary thromboprophylaxis in ambulatory cancer patients (Figure 1). The future directions provided by the recent research papers are also presented. 

## 2. Guideline Recommendations

The European Society for Medical Oncology (ESMO) 2010 practical guidelines proposed the Khrorana model to identify ambulatory cancer patients who are clinically at high risk for VTE [5]. The new 2023 ESMO guideline suggested the use of the same Khrorana model, but also the Vienna-CATS and COMPASS-CAT methods [6]. 

The same guideline strongly recommended ultrasound diagnosis and computer tomography (CT) pulmonary angiogram if deep venous thrombosis (DVT) or pulmonary embolism (PE), respectively, were suspected in cancer patients [6]. The D-dimer levels and the clinical prediction rules have not been recommended in this setting [6].

The new 2023 American Society of Clinical Oncology (ASCO) guideline considers only the Khrorana score in stratifying the risk in this setting [7]. The American Society of Hematology (ASH) 2021 guidelines provided strong and conditional recommendations for not using thromboprophylaxis on low and intermediate-risk ambulatory oncologic patients receiving cancer chemotherapy, respectively [8]. A validated risk assessment tool (i.e., Khorana score) together with clinical judgment and experience were recommended for patient classification [8]. The hereditary thrombophilia tests were suggested by the 2023 update in ambulatory cancer patients receiving systemic therapy with VTE family history determined to be at low or intermediate risk for VTE [9]. 

The European Society of Cardiology (ESC) 2022 cardio-oncology guideline recommended the baseline clinical and biomarkers assessment of the patient diagnosed with cancer [10]. Multidisciplinary monitoring during the specific treatments was also proposed [10,11]. The ESC 2022 cardio-oncology guideline emphasized the high incidence of VTE among cancer patients and recommended imaging screening in patients with clinically suspected VTE [10]. Lower-extremity venous ultrasonography is the method that has to be used in DVT diagnosis, as well as contrast-enhanced CT [10]. After detailing the patient-, cancer-, and treatment-related risk factors, the guideline proposed the TBIP method in the anticoagulation decision. It represents an acronym for thromboembolic risk, bleeding risk, drug–drug interactions, and patient preferences [10].

The ESC guideline recommended that the VTE risk assessment in ambulatory patients be individually determined and only found the Khorana and COMPASS-CAT scores useful in this setting [10]. Khrorana risk assessment was also recommended by the American College of Cardiology in ambulatory oncologic patients [12]. 

Table 1 summarizes the relevant guideline recommendations on VTE risk assessment in ambulatory oncologic patients. 

## 3. VTE Screening in Ambulatory High-Risk Oncologic Patients

Cancer and thrombosis are strongly related. VTE can be the first clinical sign of undiagnosed cancer, especially when the event is unprovoked [13], while cancer represents a risk factor in VTE occurrence. In this last setting, guidelines issued recommendations for hospitalized and surgical cancer patients and high-risk outpatients. Gainsbury et al. found a 10.1% prevalence of preoperative deep venous thrombosis (DVT) in asymptomatic patients undergoing major oncologic surgery and suggested the preoperative screening with lower extremity venous duplex ultrasound (US) in this setting [14]. Increasing age, recent diagnosis of sepsis, and a history of prior VTE were significantly associated with preoperative DVT [14]. 

Detecting VTE high-risk outpatients with cancer is still challenging. VTE screening may be an answer in this setting. In total, 6.6% of venous thrombosis was found by Heidrich et al. in all tumor patients [15]. The same authors reported a much higher incidence of 33% when using an imaging prospective approach [15]. Loftus et al. researched the role of venous US screening in incidentally detecting VTE in high-risk patients with cancer in a multicenter trial. The studied 117 patients were asymptomatic, had a Khorana score ≥ 3, and were starting new systemic chemotherapy [16]. The lower-limb venous US and a contrast-enhanced CT baseline screening discovered 9% incidental VTE (6% DVT, 1% pulmonary embolism, 1% DVT and pulmonary embolism) [16]. The patients were screened further every four weeks for a 12-week period with venous US and at 12 weeks with contrast-enhanced CT [16]. Researchers proposed the lower-limb venous US screening in addition to the oncologic surveillance CT in high-risk ambulatory cancer patients setting with a Khorana score ≥ 3 [16].

This approach could help in early VTE detection in latent stages, preventing VTE progression and thus decreasing morbidity and costs [16]. Kourlaba et al. also reported US screening of high-risk cancer patients as a cost-effective strategy compared to clinical surveillance, even when all patients with a positive first US underwent a second US [17]. Kunapareddy et al. proposed an electronic alert to identify high-risk patients and suggest US screening for early detection [18]. Holmes et al. reported the success of a multidisciplinary program related to Venous Thromboembolism Prevention in the Ambulatory Cancer Clinic (VTEPACC) [19]. The high-risk patients identified by Khorana and Protecht scores (≥3 points) were offered a hematology consultation to consider VTE prophylaxis, further referring the results of the consultation to the oncologist [19].

VTE risk was predicted by baseline D-dimer levels [20,21]. Niimi et al. recently reported the optimal D-dimer cut-off value of 4.0 μg/mL for predicting DVT in patients with malignancy [22]. Its association with risk assessment scores performed better in VTE prediction [21,22]. D-dimer was reported in another study as part of the thromboembolism risk assessment when added to fibrinogen level [23]. Oi et al. found that high D-dimer levels at VTE diagnosis were associated with an increased risk for short-term and long-term mortality and with long-term recurrent VTE, especially in patients with active cancer [24]. During a median follow-up of 30 months, D-dimer positively correlated with the reoccurrence of VTE (*p* = 0.0299) and mortality in cancer patients with VTE (*p* < 0.0001) and without VTE (*p* = 0.0008) [25]. D-dimer level positively correlated in Koch et al.’s study with VTE reoccurrence and mortality during a 30-month period [25]. The relationship with mortality was reported both in cancer patients who presented VTE and in cancer patients without VTE [25].

Another VTE risk factor is the soluble P-selectin (sP-selectin). A cut-off level of 53.1 ng/mL could predict VTE in cancer patients with no difference between tumor sites [26]. Zhang et al. recommended sP-selectin level for early identification of cancer-associated VTE and monitoring [27].

Khorana et al. recently studied the biomarkers distribution in patients with and without VTE diagnosed with cancer [28]. In the two groups, there were reported baseline lower levels of stromal cell-derived factor-1, thyroid-stimulating hormone, and monocyte chemotactic protein 4 and higher levels of growth hormone and interleukin-1 receptor type 1 [28]. ST2, IL-8, and C-reactive protein were significantly different between survivors and those who died [28]. 

Table 2 presents the relevant studies presenting modalities and importance of VTE screening among ambulatory cancer patients.

microRNAs (miRNAs) represent a promising class of biomarkers in VTE prediction in cancer, but until now, only a few small-sample-size studies, lacking external validation, have investigated their role in this setting [29]. The long non-coding RNAs (lncRNAs) may have a role as well in VTE pathogenesis [30]. Ten lncRNAs were implicated in VTE pathogenesis, but future research is needed in this setting [30].

Genetic assessment may help VTE risk stratify and prognostic in the cancer population. Thrombogenesis-related genetic polymorphisms are already studied in this setting and are integrated in specific risk scores alone, or together with clinical features. However, more prospective studies are required before clinical application. 

## 4. VTE Risk Assessment Using Scores

Because the CAT risk factors are multifactorial, risk scores have been developed to find oncologic outpatients who need anticoagulation treatment. The Khorana score was the first proposed [31]. The type of cancer, some components of the complete blood count, and body mass index were assessed. A value ≥ 2 was retained by the guidelines as describing high-risk patients [6,7,32], although a value of more than 3 was initially proposed [31]. Khorana et al. validated the method in a cohort with 34.6% breast cancer patients and 18.9% lung cancer patients [31]. The rest of Khorana et al.’s cohort had colon, ovarian, gastric, and pancreatic cancers, lymphomas, and other tumor types [31]. Mulder et al.’s meta-analysis reported the Khorana score as a tool for selecting high-risk VTE in oncologic patients [33], and Akasaka-Kihara et al. validated it in the Japanese cancer population [34]. Ramos-Esquivel et al. recently found the Khorana score to perform an accurate categorization of VTE risk in ambulatory Hispanic patients who were newly diagnosed with solid tumors and were receiving systemic chemotherapy [35]. El-Sayed et al. reported a calculated VTE occurrence probability of 87.5% when using the Khorana score at cut-off levels ≥ 3 in patients with hematological malignancy [36]. However, many researchers found the universal use of the Khorana score in primary thromboprophylaxis risk assessment inappropriate. Ha et al. (2023) only partially validated the Khorana score in the Korean population [37]. Khorana could stratify the 6-month VTE risk only in selected cancer populations [33,38,39,40]. Verzeroli et al. (2023) recently found that the Khorana score was not able to discriminate between low and high VTE risk in newly diagnosticated metastatic cancer (non-small cell lung, gastric, colorectal, and breast cancers) for whom systemic chemotherapy was indicated [41]. This score was unable to stratify VTE risk in lung cancer [39,42,43], endometrial [40], MM [44], myeloid leukemia [45], hepatocellular carcinoma [46], uterine [47], or lymphoid malignancies [48]. Although this risk score was suboptimal in VTE risk prediction, other studies found it more useful in mortality prediction [41,43] or when a value ≥ 2 was tested [49,50] in the VTE setting.

Ay et al. [51] proposed a new VTE risk assessment method in patients by adding two biomarkers, D-dimer and sP-selectin, to the Khorana score. Higher D-dimer (cut-off 1.44 μg/mL) and sP-selectin (cut-off 53.1 ng/mL) levels were reported previously by the authors to be associated with VTE [20,26] in a Vienna Cancer and Thrombosis Study (Vienna CATS). In a multinational, prospective cohort study, the Vienna CATS method discriminated better than the Khorana score between low- and high-risk VTE patients [52]. The eligible patients were those with advanced cancer who underwent chemotherapy or had started chemotherapy in the previous three months [52]. In hematological malignancy, the calculated probability of VTE occurrence was the same when using Vienna CATS or Khorana score at cut-off values of ≥3 [36]. A value more than or equal to 3 of the Vienna CATS risk score was significantly associated with VTE complications in Japanese patients with advanced cancer who were receiving chemotherapy [53]. 

Verso et al. proposed another risk score, the PROTECHT score, that added gemcitabine and platinum-based chemotherapy to Khorana score variables [54]. Moik did not sustain the use of these variables in prediction models [55]. The gemcitabine therapy has not been associated with an increased VTE risk, while platinum-based treatment had only limited predictive value beyond tumor site category and D-dimer levels [55]. However, in van Es et al.’s study, the PROTECHT score performed better discrimination than the Khorana score in the VTE risk assessment [52]. Ramos-Esquivel et al. also found PROTECHT (cut-off 3) among the scores that could categorize the VTE risk in newly diagnosed solid tumors in the Hispanic population [56]. Other studies reported this score as suboptimal in VTE risk assessment [49,50], proposing a 2-value threshold to improve the results [49].

Another risk score proposed in this setting was CONKO [57]. From the Khorana score variables, BMI was replaced by WHO performance status [57]. In Qin et al.’s (2023) study, Khorana, Vienna CATS, PROTECHT, and CONKO risk scores moderately assessed the VTE risk in hospitalized metastatic colorectal cancer inpatients [58], but the prediction was enhanced when KRAS and BRAF mutations were added to the scores [58]. In ambulatory oncologic patients, the VTE risk stratification by CONKO was suboptimal [49,52], although in the Hispanic population the results were more encouraging [56]. HYPERSCAN study reported that CONKO scores significantly stratified patients for VTE risk, while the KRS and the PROTECHT failed in ambulatory lung cancer patients [59]. In this setting, future research is awaited. Yan et al.’s systematic review regarding VTE risk assessment models for use in ambulatory patients with lung cancer is expected to be published soon [60]. 

Papinger et al. proposed a modified Vienna CATS, the CATS/MICA score, that integrated the tumor site category and D-dimer level to predict the VTE risk in ambulatory patients with solid cancers [61]. The new Vienna CATS and the CONKO scores significantly stratified patients for VTE risk in lung cancer [59]. Verzeroli et al. reported that a modified Vienna CATS score > 60 points was an independent risk factor for mortality in outpatients with metastatic cancer during chemotherapy [41].

Gerotziafas et al. proposed the COMPASS-CAT model, another VTE risk assessment method in a breast, colorectal, lung, or ovarian cancer cohort [62]. COMPASS-CAT takes into account several aspects referring to the time the cancer was diagnosed, the stage of the disease, previous VTE occurrences, platelet count, the presence of central venous catheter and the cardiovascular risk factors, the specific anti-hormonal treatment or with anthracycline, and recent hospitalization in acute medical setting [62]. The model presented in Gerotziafas et al.’s study had a good sensitivity of 88% but a lower specificity of 52% [62]. There was a strong association between catheter-related thrombosis and high Khorana, PROTECHT, and COMPASS-CAT scores [63]. COMPASS-CAT better identified more patients in high-risk group in non-small cell lung cancer [42]. Rupa-Matysek et al. reported that the COMPASS-CAT model was the most accurate predictor of VTE in lung cancer patients, compared to Khorana, PROTECHT, and CONKO scores [64]. In a large retrospective external validation study, the COMPASS-CAT model had good negative predictive value, with moderate discrimination and poor calibration power [65]. Pestana et al. (2023) reported a high VTE risk when evaluated by the COMPASS-CAT model (score ≥ 7) in breast cancer patients [66] and proposed the combination of this model with IL-10 levels to improve the method [66].

Cella et al. proposed a new assessment tool, the ONKOTEV score, an easy-to-use and cost-effective model based on clinical information, avoiding highly selective biochemical parameters. The ONKOTEV score offers one point for a Khorana score > 2, previous venous thromboembolism, metastatic disease, and vascular/lymphatic macroscopic compression [67]. In their prospective study, the area under the curve of ONKOTEV over the Khorana score was reported at 3 months (71.9% vs. 57.9%, *p* = 0.001), 6 months (75.4% vs. 58.6%, *p* < 0.001), and 12 months (69.8% vs. 58.3%, *p* = 0.014) [67]. Cella et al. (2023) validated this model in the ONKOTEV-2, a multicenter prognostic study on ambulatory patients with solid tumors undergoing active treatments [68]. The most represented tumors were breast (18.1%), gastroesophageal adenocarcinoma (16.5%), colon (12.7%), lung (11.1%), rectum (10.8%), and pancreatic cancers (7.5%) [68]. Di Nisio et al. reported that the performance of the Khorana, PROTECHT, CONKO, and ONKOTEV scores improved at the threshold of 2 points, compared to 3 points [49]. The scores’ accuracy decreased over time, suggesting the need for periodic re-evaluation [49]. An ONKOTEV score ≥ 2 was associated with a higher VTE occurrence in patients with pancreatic cancer, including ambulatory ones [69]. ONKOTEV score performed better than PROTECHT, COMPASS-CAT, CONKO, Khorana, and the CATS/MICA score in VTE risk assessment in hospitalized medical patients with primary lung cancer [70]. An ONKOTEV score ≥ 2 was also a predictor of survival and thromboembolic events in cholangiocarcinoma [71].

Table 3 summarizes the relevant studies related to the scores used in VTE risk assessment in ambulatory cancer patients. 

Zhang et al. studied the systemic immune-inflammation index (SII) and the prognosis nutritional index (PNI) in VTE prediction in gastrointestinal cancer patients [72]. The SII was an auxiliary diagnostic test for patients with venous thrombosis in general, with an AUC of 0.861 (95% CI: 0.820–0.902; *p* < 0.001), a sensitivity of 78.1%, and a specificity of 73.1% for an SII > 755.54 [73]. SII (cut-off 504.80) paired with PNI (cut-off 45.57) were part of the two nomograms proposed to predict VTE risk in gastrointestinal cancer [72]. Model A (age, tumor location, therapy, PNI, SII) and Model B (age, tumor location, therapy, PNI, SII and D dimer) presented an AUC of 0.806 (95% CI: 0.782–0.830) and 0.832 (95% CI: 0.810–0.855), respectively, as compared to Khorana score’s 0.592 (95% CI: 0.562–0.621) [72]. Zhang et al. also tested SII in lung cancer patients and developed a new nomogram model (the inflammatory marker, coagulation indicator, and tumor features) to perform an accurate prediction of VTE [74]. The SII cut-off was 851.51, and the new nomogram presented an AUC of 0.708, compared to the Khorana score’s 0.600 [74]. 

Li et al.’ new nomogram contains common data from the electronic health record, some demographic data (Asian/Pacific islander), the original Khorana score, but with cancer subtypes that were revised, and cancer and patient risk factors such as hormonal/target therapy, advanced cancer, previous VTE occurrence, recent hospitalization, and history of immobility [75]. 

Approximately 50% of cancer patients receiving modern systemic therapy were stratified into a high-risk group (a 6-month VTE risk of 8–10%) and the other half into a low-risk group (a 6-month VTE risk of 3%) [75]. This novel tool appeared generalizable in variate age, sex, and race/ethnicity subgroups but needs further validation [75].

In the ambulatory cancer population, there is no universal method for the VTE risk assessment. Thus, new specific risk scores have been developed. In newly diagnosed NSCLC outpatients who undergo chemotherapy, Thrombo-NSCLC (FVIII and sP-selectin values) predicted VTE significantly better than the Khorana score [76]. Gomez-Rosas et al. recently proposed a new risk tool, the Hypercan score, to stratify lung cancer patients for VTE and mortality risk [59]. This score contains information regarding ECOG performance and D dimer and stratifies the patients into low- and high-risk groups [59]. Using the Hypercan score, the cumulative incidence of VTE was 6% in the low- and 25% in the high-risk group [59]. Li et al. proposed and validated a new nomogram for VTE risk prediction in patients recently diagnosed with lung cancer [77]. Some clinical and therapeutic features and genetic parameters were incorporated into the new assessment system: overweight, adenocarcinoma, stage III-IV, central venous catheters, D-dimer levels ≥ 2.06 mg/L, prothrombin time ≥ 11.45 s, fibrinogen levels ≥ 3.33 g/L, triglyceride ≥ 1.37 mmol/L, ROS1 rearrangement, chemotherapy history and radiotherapy history [77]. In lymphoma patients, the ThroLy score was proposed. It was designed for both hospitalized and outpatient settings and included data referring to tumor spread (mediastinal involvement, extranodal localization), frailty (reduced mobility, BMI > 30 kg/m^2^), the presence of previous arterial or venous thromboembolic events, anemia (hemoglobin level < 100 g/L), neutropenia [78]. ThroLy not only predicted VTE in Hodgkin lymphoma, but also survival [79]. It has been studied in diffuse large B-cell lymphoma settings as well [80]. A simplified model was proposed (high-risk with a score ≥ 3 and low-risk score < 3) [80]. Others did not find this score an accurate model for predicting VTE events in patients at higher risk of VTE [81]. An adapted TiC-Onco risk score to lymphoma settings was proposed by Bastos-Oreiro et al. [82] with promising results. The TiC-LYMPHO score incorporated the same genetic variables included in the TiC-ONCO score and some of the clinical variables associated with VTE in the studied population (the type of lymphoma according to the WHO classification, mediastinal involvement, Ann Arbor stage, bed rest for >3 days, and a family or personal history of VTE) [82]. López Sacerio et al. (2023) found five predictive factors—hypercholesterolemia, tumoral activity, use of thrombogenic drugs, diabetes mellitus, and immobilization—that were integrated into a predictor model of VTE in patients with hospitalized hematologic malignancies [83]. 

The IMPEDE VTE score (immunomodulatory agent; body mass index ≥ 25 kg/m^2^; pelvic, hip or femur fracture; erythropoietin stimulating agent; dexamethasone/doxorubicin; Asian ethnicity/race; VTE history; tunneled line/central venous catheter; existing thromboprophylaxis) was developed and validated by Sanfilippo et al. (2019) as a VTE risk prediction score in multiple myeloma (MM) [84]. Recently, Sanfilippo et al. (2023) reported that adding D-dimer to the IMPEDE VTE score could improve VTE prediction among MM patients [85].

The PICOS score (primary tumors with high thrombogenicity, immobilization, chemotherapy, obesity, and steroid) was proposed by Wolpert et al. as a helpful tool for the identification of patients with brain metastasis at high risk for VTE [86].

Kubo et al. found that the D-dimer combined with the Glasgow prognostic score accurately predicted VTE in stage IIIC and IVA of ovarian cancer (AUC: 0.846; *p* < 0.001) [87]. D-dimer could significantly predict VTE in all gynecologic cancer patients. Optimal reported D-dimer cut-off values were 3.1, 3.2, and 3.9 μg/mL in cervical, endometrial, and ovarian cancer patients, respectively [87].

## 5. Genetic-Based Risk Assessment Scores

Polygenic risk scores do not change during the cancer course. Thus, they could be potential predictors of cancer-associated VTE independent of cancer type. Both Factor V Leiden and ABO gene mutations were reported as independent predictors of VTE occurrence in moderate to high-risk outpatients with cancer undergoing chemotherapy [88].

Lindström et al. (2019) reported the results of a large genome-wide association study (GWAS) and the first transcriptome-wide association study (TWAS) on VTE risk. GWAS meta-analysis identified 34 independent genetic signals for VTE risk with 14 newly reported associations [89]. TWAS identified five additional genetic loci not previously associated with VTE (SPSB1, ERAP1, RP11-747H7.3, RP4-737E23.2, and replicated SH2B3) [89]. The researchers demonstrated that a genetic risk score based on 37 VTE-susceptibility variants can identify a subset of the population at high risk for developing VTE [89].

De Haan et al. designed a genetic score to select VTE high-risk patients [90]. This method contained data on 31 single-nucleotide polymorphisms (SNPs) associated with an increased risk [90]. The 5-SNP score (rs8176719, rs6025, rs1799963, rs2066865, and rs2036914) was created by adding one-by-one the SNPs with the highest odds ratios of VTE and similarly discriminated high-risk patients as 31 SNPs regarding both incidental and recurrent VTE events [91]. VTE risk increased with the number of prothrombotic risk alleles, independent of the cancer diagnosis [91]. The presence of both prothrombotic risk alleles and cancer represented a highly elevated VTE risk factor [91].

Other studies did not find the 5-SNP score to be superior to the Khorana score [50]. Jakobsen et al. reported an elevated discriminative effect of the 5-SNP score on VTE risk by combining it with the mean platelet volume, but the results were not focused on the cancer population [92]. 

The 5, 37, 297, extended 297, and 100 SNPs prospectively identified those cancer patients at high risk for VTE development in a population-based study [93]. The tumor type has not influenced the result [93]. The 36,150 patients of the UK Biobank cohort diagnosed with hematological or solid cancer were studied from the genetic point of view regarding VTE risk. In the 12 months post-cancer diagnosis, the germline genetic markers accurately selected the patients with an increased double risk for VTE occurrence. Guman et al. demonstrated that the tumor type and polygenic scores’ performance were independent variables, as the latter remained consistent during the 12-month follow-up [93]. The VTE prediction was improved when the two variables were combined [93].

The TiC-Onco risk score integrated genetic (rs2232698, rs6025, rs5985, rs4524) and clinical risk factors. Muñoz et al. used it to identify patients with colorectal, esophagogastric, lung, or pancreatic cancer in the outpatient setting who are at high risk of VTE [94]. This method has to be followed at the moment cancer is suspected [94]. Its sensitivity was significantly higher than that of the Khorana, while the specificities of both scores were similar in the studied population [94]. TiC-LYMPHO, a modified TiC-Onco score, was proposed in lymphoma settings [82]. Neto et al. (2023) reported PROCR rs10747514 and RGS7 rs2502448 as valuable prognostic biomarkers regardless of VTE and significantly associated with the VTE risk in cervical cancer [95]. 

Recently, nine genetic variants (rs4524, rs6025, rs2232698, rs2227631, rs268, rs169713, rs11696364, rs5110, rs6003) were independently associated with VTE in outpatients with cancer [96]. Muñoz et al. developed and validated ONCOTHROMB, by combining this genetic profile with three clinical variables independently associated with VTE in outpatients with cancer. This score, with a higher AUC compared to the Khorana model (AUC, 0.781 vs. 0.580; *p* < 0.001) was recommended to be assessed at the moment cancer is suspected [96]. ONCOTHROMB presented a significantly higher sensitivity than Khorana (81.54% vs. 22.54%; *p* < 0.001), with a lower specificity (65.22% vs. 81.76%; *p* < 0.0001) [96].

## 6. Machine Learning Algorithms Tools

Artificial intelligence brings new methods to assess risk. Machine learning (ML) can develop many statistical algorithms that can learn from the pattern of the database. The highly flexible novel tools can discriminate better in a nonlinear setting [97,98]. The manual data analysis is eliminated when using ML algorithms, while a large volume of data can be reviewed, identifying more easily patterns and trends. The automatic and dynamic self-learning process leads to continuous improvement in decision making. ML may help save time and resources at the clinician level by reducing the data analysis time, optimizing medical decisions, and offering insights into other centers’ experiences and databases. Thus, the clinician may have additional time to spend with patients to understand better their needs and disease. Of course, the input data must be correct and large enough to obtain accurate results. The parameters and ML algorithms must be continuously developed and optimized. Otherwise, the probability of high errors is high. The users must define the acceptable margins of the statistical error because the ML algorithms approach represents a probabilistic process. The ethical challenges in collecting and handling data represent another ML algorithm limitation. Clinicians must understand the advantages and the disadvantages of using the ML approach as they are the interface to the patients [99]. Clinical judgment still has its role in our era. The medical doctor is the one able to discriminate between clinical changes by integrating medical and social data. Still, the clinician is the one who can interpret the results of ML algorithms in the patient’s context. 

Machine learning algorithms can help assess the risk of venous thromboembolism (VTE) in ambulatory cancer patients. By accurately identifying high-risk VTE patients, healthcare professionals can provide them with the appropriate treatment. In Ferroni et al.’s (2017) study, a model based on multiple kernel learning (MKL) and random optimization (RO) was used to achieve this goal in chemotherapy-treated ambulatory cancer patients [100]. ML-RO-2 was the most accurate model compared to the Khorana score (positive likelihood ratio 1.68, negative likelihood ratio 0.24) [100]. ML-RO-2 presented an area under the precision–recall curve of 0.212, while the Khorana score’ area was only 0.096 [100]. The strongest association was related to the blood lipids and body mass index/performance status, while the weaker was related to the tumor site/stage and drugs [100]. The second best-performing model was ML-RO-3 [100]. A study conducted by Ferroni et al. in 2017 validated the ML-RO-2 and ML-RO-3 approaches as a low-cost method for assessing VTE risk in oncologic patients [101]. The f-measure, a metric used in ML, calculated as a harmonic mean of *P* (positive predictive value in ML) and *R* (sensitivity in ML), measured the effectiveness of a classifier algorithm. ML-RO-2 and ML-RO-3 presented higher f-measures (0.213 and, respectively, 0.211) than the Khorana score (f-measure: 0.100) [101]. The study involved 608 patients, with a mean age of 63 years, with 58% of them having relapsing/metastatic solid cancers [101]. The incidence of deep venous thrombosis was 5.3%, while pulmonary embolism was diagnosed in 1.8% of cases [101]. Xu et al. (2023) developed and validated a new clinical prediction model for VTE in gastric cancer patients based on support vector machine (SVM), one of the ML algorithms [97]. The model’s AUC, sensitivity, and specificity were 0.825, 0.710, and 0.802, respectively [97]. The top five predictors in the model were the clinical stage, the blood transfusion history, D-dimer, age, and fibrinogen degradation products [97]. 

Jin et al. reported that only linear discriminant analysis (AUC 0.773) and logistic regression (AUC 0.772) outperformed the Khorana score (AUC 0.642) in cancer-related VTE prediction [102]. The combination with D-dimer improved the models’ performance [103]. The top five predictors of cancer-related VTE were D-dimer level, age, Charlson Comorbidity Index, length of stay, and previous VTE history [102].

Lei et al. recommended the random forest model as the best classifier for VTE prediction in lung cancer [103]. The model presented an AUC of 0.91 (95% CI: 0.893–0.926), a sensitivity of 0.714 (95% CI: 0.614–0.762), and a specificity of 0.965 (95% CI: 0.941–0.985) [103]. The five most relevant parameters were Karnofsky Performance Status, a history of VTE, recombinant human endostatin, EGFR-TKI, and platelet count [103]. 

Mantha et al. (Preprint) [104] conducted the first study of a deep-learning model that predicts the risk of cancer-associated VTE. The model was selected based on its C-index and potential usefulness in clinical practice [104]. The DeepHit model’s most important predictors were plasma albumin, followed by the presence of metastatic disease [104]. Additionally, the use of systemic therapy, plasma electrolytes (sodium, potassium, chloride, and calcium), hemoglobin, glucose, and alkaline phosphatase were identified as VTE risk predictors [104]. Meng et al. found in hospitalized cancer patients that the extreme gradient boosting (XGBoost) model achieved the best performance in VTE prediction [105]. The five most significant features tested in the model were D-dimer level, diabetes, hypertension, pleural metastasis, and hematological malignancies [105]. Danilatou and colleagues demonstrated that the machine learning approach outperformed traditional scoring systems in predicting early and late mortality in critically ill patients with venous thromboembolism and cancer. In addition, they validated the model externally [106].

In the future, machine learning models have potential in clinical practice but require optimization with larger databases and multiple algorithms. It is crucial to adhere to the standardization of reporting provided by the Scientific and Standardization Committee (SSC) Subcommittee on Hemostasis & Malignancy of the International Society on Thrombosis and Hemostasis (ISTH). This involves the TRIPOD checklist (Transparent Reporting of a Multivariable Prediction Model for Individual Prognosis or Diagnosis), clearly defining predictors, defining the derivation population, and validating the model externally before implementing it [107].

This paper aims to present the current relevant knowledge in VTE risk assessment in ambulatory cancer patients, starting from the guideline recommendations and continuing with the specific risk assessment methods and machine learning models approaches. The main limitation of this review is its narrative character.

## 7. Conclusions and Future Directions

VTE risk assessment in ambulatory cancer patients is still challenging. High-risk cancer patients must be accurately discriminated against thromboprophylaxis, but the guidelines do not provide enough information. Many scores, nomograms, and models were developed, but none have optimally performed in this setting. Clinical features, biomarkers, and genetic patterns have been tested alone or grouped in cancer populations in general or specific cancer cohorts. The polygenic risk scores that do not change during the cancer course could be potential predictors of cancer-associated VTE independent of cancer type, but this idea needs further validation in prospective studies. Additionally, the expenses must be assessed better. The machine learning models might provide a potentially useful algorithm through learning and improving its performance based on the data they use. But, to apply these methods in clinical practice, they need to be optimized in larger databases. 

## Figures and Tables

**Figure 1 cancers-16-00458-f001:**
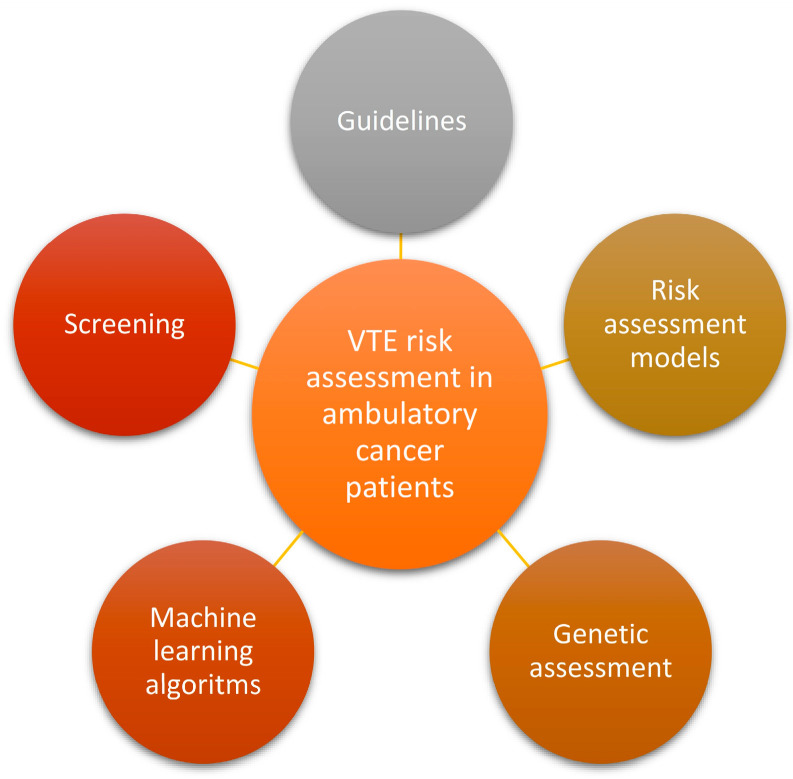
The current relevant knowledge in VTE risk assessment in ambulatory cancer patients. Abbreviations: VTE—venous thromboembolism.

**Table 1 cancers-16-00458-t001:** The relevant guideline recommendations on VTE risk assessment in ambulatory oncologic patients.

Guideline	Reference	Main Findings	Recommendation
ESMO 2023	[6]	Ultrasound diagnosis in suspected DVT and diagnosis by CTPA in suspected PE, without using clinical prediction rules and D-dimer level	Class I, Level of evidence A
		Apixaban, rivaroxaban or LMWH may be considered for primary thromboprophylaxis for a maximum of 6 months in high-thrombosis-risk ambulatory cancer patients starting systemic anticancer treatment.	Class I, Level of evidence B
		Primary thromboprophylaxis is suggested when a VTE risk is estimated to be >8–10% at 6 months.	Class II, Level of evidence C
		Cancer patients should be offered a CAT risk assessment and have an opportunity to discuss their particular risks.	Class III, Level of evidence B
		Khrorana score (cut-off 2), COMPASS-CAT, and Vienna-CATS should be used for risk stratification	Class III, Level of evidence C
		LMWH given at a higher dose for a maximum of 3 months may be considered for ambulatory pancreatic cancer patients on first-line systemic anticancer treatment,.	Class II, Level of evidence C
ASCO 2023	[7]	Routine pharmacologic thromboprophylaxis should not be offered to all outpatients with cancer	Evidence-based Intermediate-High quality Strong recommendation
		High-risk outpatients with cancer (Khorana score ≥ 2 before starting a new systemic chemotherapy regimen) may be offered thromboprophylaxis with apixaban, rivaroxaban, or LMWH provided there are no significant risk factors for bleeding and no drug interactions. Such therapy should be accompanied by a discussion with the patient about the relative benefits and harms, drug cost, and duration of prophylaxis in this setting	Evidence-based Evidence quality: Intermediate to High for apixaban and rivaroxaban, Intermediate for LMWH Moderate recommendation
		Patients with multiple myeloma receiving thalidomide- or lenalidomide-based regimens with chemotherapy and/or dexamethasone should be offered pharmacologic thromboprophylaxis with either aspirin or LMWH for lower-risk patients and LMWH for higher-risk patients	Evidence-based Intermediate evidence quality Strong recommendation
ESC 2022	[10]	The venous ultrasonography or contrast-enhanced CT were recommended as screening when clinical signs of DVT were present	
		The CT pulmonary angiography was recommended as screening when clinical signs of PE are present	
		TBIP assessment	Class I
		VTE risk should be individually determined (Khorana score or COMPASS-CAT)	-
		For ambulatory patients with cancer at high risk of thrombosis receiving systemic therapy, primary thromboprophylaxis with a NOAC (apixaban or rivaroxaban) or LMWH may be considered, provided there are no significant contraindications.	Class IIb, Level of evidence B
		The patients at high risk of thrombosis receiving systemic therapy are those with locally advanced/metastatic pancreas or lung cancer or Khorana score ≥ 2	
		A discussion with the patient about the relative benefits and harms, cancer prognosis, drug cost, and duration of treatment is recommended prior to prophylactic anticoagulation for the primary prevention of VTE	Class I, Level of evidence C
ASH 2021	[8]	For low-risk thrombosis patients receiving systemic therapy, no thromboprophylaxis is recommended over parenteral thromboprophylaxis	Strong recommendation, moderate certainty in the evidence of effects
		For intermediate-risk thrombosis patients receiving systemic therapy, no prophylaxis is suggested over parenteral prophylaxis	Conditional recommendation, moderate certainty in the evidence of effects
		For high-risk thrombosis patients receiving systemic therapy thromboprophylaxis (LMWH or DOAC) are suggested over no thromboprophylaxis Classification of patients as being low-, intermediate-, or high-risk for VTE should be based on a validated risk assessment tool (i.e., Khorana score) complemented by clinical judgment and experience.	Conditional recommendation, moderate certainty in the evidence of effects

Abbreviations: ASCO, American Society of Clinical Oncology; ASH American Society of Hematology; COMPASS-CAT, Comparison of Methods for Thromboembolic Risk Assessment with Clinical Perceptions and Awareness in real-life patients-Cancer-Associated Thrombosis; CAT, cancer-associated thrombosis; CT, Computed tomography; CTPA, CT pulmonary angiogram; ESC, European Society of Cardiology; DOAC, direct oral anticoagulants; DVT, deep venous thrombosis; ESMO, European Society for Medical Oncology; LMWH, low molecular weight heparin; PE, pulmonary embolism; TBIP, thromboembolic risk, bleeding risk, drug–drug interactions, patient preferences; Vienna-CATS, Vienna Cancer and Thrombosis Study; VTE, venous thromboembolism.

**Table 2 cancers-16-00458-t002:** The relevant studies presenting modalities and importance of VTE screening among ambulatory cancer patients.

Screening Modality	Authors (Year) [Ref]	No. Patients	VTE Detected (%)	Type of Tumors	Main Findings
Lower limb venous duplex US	Gainsbury et al. (2018) [14]	346	10.1	Solid cancer	High-risk cancer patients may benefit from screening lower extremity venous duplex US before surgery.
Lower limb duplex US and/or venography	Heidrich et al. (2009) [15]	97	33	Various types	Regular screening for thrombosis is indicated even in asymptomatic tumor patients
Lower limb duplex US and contrast-enhanced chest CT	Loftus et al. (2022) [16]	117	58	Solid cancers	Suggested to add US screening to routine oncologic surveillance CT in high-risk ambulatory cancer patients (Khorana score ≥ 3)
Lower limb venous US	Kourlaba et al. (2017) [17]	907	-	various	Screening high-risk cancer patients via US to detect asymptomatic DVT is a cost-effective strategy over clinical surveillance
Automated alert Lower limb venous US	Kunapareddy et al. (2019) [18]	194	12.5	various	An automated alert may help in early detection of DVT in high-risk cancer patients
VTEPACC model	Holmes et al. (2020) [19]	918	23.2	various	VTEPACC involves a multidisciplinary approach
D-dimer F 1 + 2	Ay et al. (2009) [20]	821	7.6	various	The cumulative probability of developing VTE after 6 months was highest in patients with both elevated D-dimer and elevated F 1 + 2
Baseline D-dimer	Schorling et al. (2020) [21]	100	11.2	Solid cancers	VTE risk was well predicted by baseline D-dimer levels.
D-dimer	Niim et al. (2023 [22]	208	28.4	various	The optimal D-dimer cut-off value for the DVT diagnosis in cancer patients was 4.0 μg/mL.
D-dimer	Oi et al. (2020) [24]	2852		various	Elevated levels at diagnosis were associated with an increased risk for short-term and long-term mortality.
D-dimer	Koch et al. (2023) [25]	526	39.73	various	Levels above the 10-fold upper reference limit contain diagnostic and prognostic information
sP-selectin	Ay et al. (2008) [26]	687	6.4	various	Higher levels independently predict VTE in cancer patients
sP-selectin	Zhang (2023) [27]	1882	24.17	various	Metaanalysis. Role in early identification and monitoring A higher level in Asian cancer patients
Various biomarkers	Khorana (2022) [28]	124	50	various	SDF-1 and TSH were the strongest predictors of VTE

Abbreviations: CT, computed tomography; DVT, deep venous thrombosis; F 1 + 2, prothrombin fragment 1 + 2; SDF-1, stromal cell-derived factor1; VTE, venous thromboembolism; VTEPACC, Venous Thromboembolism Prevention in the Ambulatory Cancer Clinic; TSH, thyroid-stimulating hormone; US, ultrasonography.

**Table 3 cancers-16-00458-t003:** The relevant studies related to the main risk scores used for selecting high-risk outpatients with cancer who would benefit from primary thromboprophylaxis.

Score	Authors (Year) [Reference]	Study Population							Observation
		No.	Type of Cancer	Age	Male (%)	Ethnicity/ Race	Metastasis (%)	VTE (%)	
Khorana	Khorana et al. (2008) [31]	2801	breast, lung, ovarian sarcoma colon lymphomas		32.7	US	36.9	2.2	Validation in an independent cohort (1365 patients). Cut-off KRS = 3
	Austin et al. (2019) [40]	87	pancreatic	66.2	-	UK	86.2	26.8	Retrospective KRS was associated with VTE in endometrial cancer only
		154	endometrial	67.5			27.3	5.7	
		205	colorectal	64			16.6	9.8	
		193	ovarian	60.2			67.9	10.2	
		91	cervical	48.9			0	0	
	Mulder et al. (2019) [33]	34,555	various	-	-	various	-	6.9%	Meta-analysis At cut-off 2, KRS helps to select high-risk patients, but with limitations in lung and hematologic cancers
	Di Nisio et al. (2019) [49]	770	various types	-	-	Multinational	70	-	KRS performance improved when using the threshold of 2 points
	van Es et al. (2020) [39]	3293	solid cancers	61	59	various	68	-	Meta-analysis KRS did not stratify the VTE risk in lung cancer patients
	Akasaka-Kihara et al. (2021) [34]	27,687	various	67	52.3	Japanese	23.5	5.26	External validation of KRS Cut-off KRS = 2 Cut-off for BMI = 25
	Guman et al. (2021) [50]	2729	advanced solid tumors	63	51	Dutch	-	5.9	Retrospective multicentre study Poor overall discrimination of KRS, PROTECHT, 5-SNP
	Ramos-Esquivel et al. (2022) [35]	708	solid tumors	59.04	37.4	Hispanic	-	4.23	Support KRS use in Hispanic patients
	Overvad et al. (2022) [38]	40,218	various	65	44.6	Danish	-	2.5	KRS did not stratify the risk of VTE in all cancer types.
	Verzeroli et al. (2023) [41]	1286	NSCL, colorectal, gastric, breast	65	55	Caucasian	100	9.7	KRS did not discriminate between low and high VTE At cut-off levels ≥ 3, independently predicted mortality
	El-Sayed et al. (2023) [36]	81	hematology	42.6	49.4	Egyptian	2.7	9.8	At cut-off levels ≥ 3, the calculated VTE probability was 87.5%
	Ha et al. (2023) [37]	11,714	various	59	40.5	East Asian	-	1.77	Partially validated KRS in Korean cancer patients
PROTECHT	van Es et al. (2017) [52]	876	solid advanced cancers	64	59	Dutch Italian French Mexican	66	6.1	Multinational prospective study Vienna CATS and PROTECHT predicted better than KRS the VTE occurrence.
	Di Nisio et al. (2019) [49]	770	various types	-	-	Multinational	70	-	PROTECHT performance improved when using the threshold of 2 points
	Guman et al. (2021) [50]	2729	advanced solid tumors	63	51	Dutch	-	5.9	Retrospective multicenter study Poor overall discrimination of KRS, PROTECHT, 5-SNP
	Ramos-Esquivel et al. (2023) [56]	708	solid tumours	-	-	Hispanic	-	4.45	Poor overall discriminatory performance for predicting all patients at VTE risk.
ONKOTEV	Cella et al. (2017) [67]	843	various types	59	33.6	Italian, Germany	55.2	8.6	Prospective ONKOTEV score was proposed
	Godinho et al. (2020) [69]	165	pancreatic	73	54.5	Portuguese	55.8	30.9	Retrospective ONKOTEV score ≥ 2 stratifies VTE risk in pancreatic cancer
	Cella et al. (2023) [68]	425	various types	61	43.1	Italian, Germany, UK	68	2.6	External validation ONKOTEV
COMPASS-CAT	Gerotziafas et al. (2016) [61]	1023	breast colorectal lung ovarian	55	18.9	Multinational	39.6	8.5	COMPASS-CAT proposal
	Spyropoulos et al. (2020) [65]	3814	breast lung colorectal ovarian	64	21	US	18.8	5.85	External validation of COMPASS-CAT
	Abdel-Razeq et al. (2023) [42]	508	NSCLC	58.4	79.7	Jordanian	65.6	15	retrospective COMPASS-CAT better identified high-risk VTE patients compared to KRS
Vienna CATS	Ay et al. (2010) [51]	819	various types	62	55.44%	Austrian	37.1	7.4	Proposed to add D-dimer and sP-selectin to KRS assessment
	van Es et al. (2017) [52]	876	solid advanced cancers	64	59	Dutch Italian French Mexican	66	6.1	Multinational, prospective study. Vienna CATS and PROTECHT predicted better than KRS the VTE occurrence.
	Harada et al. (2023) [53]	190	solid cancers	69	73	Japanese	100	8.94	single-center, prospective study unresectable cancer patients levels ≥ 3 were significantly associated with VTE occurrence
	El-Sayed et al. (2023) [36]	81	hematology	42.6	49.4	Egyptian	2.7	9.8	Prospective study At cut-off levels ≥ 3, the VTE occurrence calculated probability was 87.5%
CATS/ MICA	Pabinger et al. (2018) [61]	1423 CATS	solid cancers	62.9 CATS	54.2 CATS	Austrian Dutch Franch Italian Mexican	61.7	6.3	The data from the CATS cohort (1423) were used to select variables. The score was externally validated in the MICA cohort (832).
		832 MICA		63.7 MICA	57.3 MICA				
	Verzeroli et al. (2023) [41]	1286	NSCL, colorectal, gastric, breast	65	55	Caucasian	100	9.7	The cut-off 60 assessed better than KRS the VTE risk The levels ≥ 60, independently predicted mortality

The missing values were either unavailable or non-applicable. Abbreviations: CATS, Vienna Cancer and Thrombosis Study cohort; COMPASS-CAT, COMPASS-CAT, Comparison of Methods for Thromboembolic Risk Assessment with Clinical Perceptions and Awareness in real-life patients-Cancer-Associated Thrombosis; MICA, Multinational Cohort Study to Identify Cancer Patients at High Risk of Venous Thromboembolism; KRS, Khorana score; No. number; NSCL, non-small cell lung; Vienna-CATS, Vienna Cancer and Thrombosis Study; VTE, venous thromboembolism; UK, United Kingdom; US, United States.

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
