# Peer review of "Novel Insights in Venous Thromboembolism Risk Assessment Methods in Ambulatory Cancer Patients: From the Guidelines to Clinical Practice"

_cancers, 2024, doi:10.3390/cancers16020458_

Round 1
Reviewer 1 Report
Comments and Suggestions for Authors
The paper by Drăgan A and Drăgan AS is a nice review of the letest development and recommendations of updated guidelines.
The review is sound and quite updated, but I find the Artificial Intelligence section a bit too short, while it should be emphasized in the light of the ever growing progresses. Some ML approaches to VTE in cancer patients should be added and discussed in terms of pros and cons. Otherwise the manuscript remains a nice review on already published issues, including the following.
Hereafter are only inidcative examples:
- Mantha S, et al. doi: 10.21203/rs.3.rs-2870367/v1
- Danilatou V, doi: 10.3390/ijms23137132
- Ferroni P, doi: 10.1155/2017/8781379
Author Response
Bucharest, January, 7th, 2023
To:
Reviewer 1
Cancers
Dear Professor,
We are submitting the revised version of our manuscript entitled “"Novel insights in venous thromboembolism risk assessment methods in ambulatory cancer patients: from the guidelines to clinical practice” to be considered for publication in Cancers as a Review. The initial version of our manuscript has been revised taking into consideration your comments.
Each point made by you is addressed below separately, explaining what we have changed and where the changes are to be found in the manuscript (the changes are highlighted in the revised manuscript).
We thank you for your useful suggestions and comments and your kind words about our manuscript. We greatly appreciate the time and efforts you have taken for the evaluation of our work!
Comments and Suggestions for Authors
The paper by Drăgan A and Drăgan AS is a nice review of the letest development and recommendations of updated guidelines.
The review is sound and quite updated, but I find the Artificial Intelligence section a bit too short, while it should be emphasized in the light of the ever growing progresses. Some ML approaches to VTE in cancer patients should be added and discussed in terms of pros and cons. Otherwise the manuscript remains a nice review on already published issues, including the following.
Hereafter are only inidcative examples:
- Mantha S, et al. doi: 10.21203/rs.3.rs-2870367/v1
- Danilatou V, doi: 10.3390/ijms23137132
- Ferroni P, doi: 10.1155/2017/8781379
Response
We thank you for your appreciation!
According to your suggestions, we have improved the Machine learning section by discussing the advantages and disadvantages of these methods and updated it with new references.
The revised manuscript includes some changes that are highlighted.
We have added a figure that summarizes the facts exposed in the review at the suggestions of Reviewer 3. We have also improved Table 1 by including the class and level of evidence for the guidelines’ recommendations. Additionally, we have added a new Table 2, which presents the relevant studies on the modalities and importance of VTE screening among ambulatory cancer patients, as suggested by Reviewer 3. We restructured Table 2 from the initial manuscript based on Reviewer 3 suggestions. The new Table 3 is more informative and includes data on the studied population, VTE incidence, main findings, and validation data.
We attach the revised section and the related references.
- Machine learning algorithms tools
Artificial intelligence brings new methods to assess risk. Machine learning (ML) can develop many statistical algorithms that can learn from the pattern of the database. The highly flexible novel tools can discriminate better in a nonlinear setting [98,99]. The manual data analysis is eliminated when using ML algorithms, while a large volume of data can be reviewed, identifying more easily patterns and trends. The automatic and dynamic self-learning process leads to continuous improvement of decision-making. ML may help save time and resources at the clinician level by reducing the data analysis time, optimizing medical decisions, and offering insights into other centers’ experiences and databases. Thus, the clinician may have additional time to spend with patients to understand better their needs and disease. Of course, the input data must be correct and large enough to obtain accurate results. The parameters and ML algorithms must be continuously developed and optimized. Otherwise, the probability of high errors is high. The users must define the acceptable margins of the statistical error because the ML algorithms approach represents a probabilistic process. The ethical challenges in collecting and handling data represent another ML algorithm limitation. Clinicians must understand the advantages and the disadvantages of using the ML approach as they are the interface to the patients [100]. Clinical judgment still has its role in our era. The medical doctor is the one able to discriminate between clinical changes by integrating medical and social data. Still, the clinician is the one who can interpret the results of ML algorithms in the patient’s context.
Machine learning algorithms can help assess the risk of venous thromboembolism (VTE) in ambulatory cancer patients. By accurately identifying high-risk VTE patients, healthcare professionals can provide them with the appropriate treatment. In the Ferroni et al. (2017) study, a model based on multiple kernel learning (MKL) and random optimization (RO) was used to achieve this goal in chemotherapy-treated ambulatory cancer patients [101]. ML-RO-2 was the most accurate model compared to the Khorana score (positive likelihood ratio 1.68, negative likelihood ratio 0.24) [101]. ML-RO-2 presented an area under the Precision-Recall curve of 0.212, while the Khorana score' area was only 0.096 [101]. The strongest association was related to the blood lipids and body mass index/performance status, while the weaker was related to the tumor site/stage and drugs [101]. The second best-performing model was ML-RO-3 [101]. A study conducted by Ferroni et al. in 2017 validated the ML-RO-2 and ML-RO-3 approaches as a low-cost method for assessing VTE risk in oncologic patients [102]. The f-measure, a metric used in ML, calculated as a harmonic mean of P(positive predictive value in ML) and R (sensitivity in ML), measured the effectiveness of a classifier algorithm. ML-RO-2 and ML-RO-3 presented higher f-measures (0.213 and, respectively, 0.211) than Khorana score (f-measure: 0.100) [102]. The study involved 608 patients, with a mean age of 63 years, with 58% of them having relapsing/metastatic solid cancers [102]. The incidence of deep venous thrombosis was 5.3%, while pulmonary embolism was diagnosed in 1.8% of cases [102]. Xu et al. (2023) developed and validated a new clinical prediction model for VTE in gastric cancer patients based on support vector machine (SVM), one of the ML algorithms [98]. The model’s AUC, sensitivity, and specificity were 0.825, 0.710, and, respectively, 0.802 [98]. The top 5 predictors in the model were the clinical stage, the blood transfusion history, D-dimer, age, and fibrinogen degradation products [98].
Jin et al. reported that only linear discriminant analysis (AUC 0.773) and logistic regression (AUC 0.772) outperformed the Khorana score (AUC 0.642) in cancer-related VTE prediction [103]. The combination with D-dimer improved the models’ performance [103]. The top five predictors of cancer-related VTE were D-dimer level, age, Charlson Comorbidity Index, length of stay, and previous VTE history [103].
Lei et al. recommended the Random Forest model as the best classifier for VTE prediction in lung cancer [104]. The model presented an AUC of 0.91 (95% CI: 0.893–0.926), a sensitivity of 0.714 (95% CI: 0.614–0.762), and a specificity of 0.965 (95% CI: 0.941–0.985) [104]. The five most relevant parameters were Karnofsky Performance Status, a history of VTE, recombinant human endostatin, EGFR-TKI, and platelet count [104].
Mantha et al. (Preprint) [105] conducted the first study of a deep-learning model that predicts the risk of cancer-associated VTE. The model was selected based on its C-index and potential usefulness in clinical practice [105]. The DeepHit model's most important predictors were plasma albumin, followed by the presence of metastatic disease [105]. Additionally, the use of systemic therapy, plasma electrolytes (sodium, potassium, chloride, and calcium), hemoglobin, glucose, and alkaline phosphatase were identified as VTE risk predictors [105]. Meng et al. found in hospitalized cancer patients that the extreme gradient boosting (XGBoost) model achieved the best performance in VTE prediction [106]. The five most significant features tested in the model were D-dimer level, diabetes, hypertension, pleural metastasis, and hematological malignancies [106]. Danilatou and colleagues demonstrated that the machine learning approach outperformed traditional scoring systems in predicting early and late mortality in critically ill patients with venous thromboembolism and cancer. In addition, they validated the model externally [107].
In the future, machine learning models have potential in clinical practice but require optimization with larger databases and multiple algorithms. It's crucial to adhere to the standardization of reporting provided by the Scientific and Standardization Committee (SSC) Subcommittee on Hemostasis & Malignancy of the International Society on Thrombosis and Hemostasis (ISTH). This involves the TRIPOD checklist (Transparent Reporting of a Multivariable Prediction Model for Individual Prognosis or Diagnosis), clearly defining predictors, defining the derivation population, and validating the model externally before implementing it [108].
- Xu, Q.; Lei, H.; Li, X.; Li, F.; Shi, H.; Wang, G.; Sun, A.; Wang, Y.; Peng, B. Machine learning predicts cancer-associated venous thromboembolism using clinically available variables in gastric cancer patients. Heliyon. 2023, 9, e12681.
- Nudel, J.; Bishara, A.M.; de Geus, S.W.L.; Patil, P.; Srinivasan, J.; Hess, D.T.; Woodson, J. Development and validation of machine learning models to predict gastrointestinal leak and venous thromboembolism after weight loss surgery: an analysis of the MBSAQIP database. Surg Endosc. 2021, 35, 182-191.
- Ting Sim, J.Z.; Fong, Q.W.; Huang, W.; Tan, C.H. Machine learning in medicine: what clinicians should know. Singapore Med J. 2023, 64, 91-97.
- Ferroni, P.; Zanzotto, F.M.; Scarpato, N.; Riondino, S.; Nanni, U.; Roselli, M.; Guadagni, F. Risk Assessment for Venous Thromboembolism in Chemotherapy-Treated Ambulatory Cancer Patients. Med Decis Making. 2017, 37, 234-242.
- Ferroni, P.; Zanzotto, F.M.; Scarpato, N.; Riondino, S.; Guadagni, F.; Roselli, M. Validation of a Machine Learning Approach for Venous Thromboembolism Risk Prediction in Oncology. Dis Markers. 2017, 2017, 8781379.
- Jin, S.; Qin, D.; Liang, B.S.; Zhang, L.C.; Wei, X.X.; Wang, Y.J.; Zhuang, B.; Zhang, T.; Yang, Z.P.; Cao, Y.W.; Jin, S.L.; Yang, P.; Jiang, B.; Rao, B.Q.; Shi, H.P.; Lu, Q. Machine learning predicts cancer-associated deep vein thrombosis using clinically available variables. Int J Med Inform. 2022, 161, 104733
- Lei, H.; Zhang, M.; Wu, Z.; Liu, C.; Li, X.; Zhou, W.; Long, B.; Ma, J.; Zhang, H.; Wang, Y.; Wang, G.; Gong, M.; Hong, N.; Liu, H.; Wu, Y. Development and Validation of a Risk Prediction Model for Venous Thromboembolism in Lung Cancer Patients Using Machine Learning. Front Cardiovasc Med. 2022, 9, 845210
- Mantha, S.; Chatterjee, S.; Singh, R.; Cadley, J.; Poon, C.; Chatterjee, A.; Kelly, D.; Sterpi, M.; Soff, G.; Zwicker, J.; Soria, J.; Ruiz, M.; Muñoz, A.; Arcila, M. Application of Machine Learning to the Prediction of Cancer-Associated Venous Thromboembolism. Res Sq [Preprint]. 2023, rs.3.rs, 2870367
- Meng, L.; Wei, T.; Fan, R.; Su, H.; Liu, J.; Wang, L.; Huang, X.; Qi, Y.; Li, X. Development and validation of a machine learning model to predict venous thromboembolism among hospitalized cancer patients. Asia Pac J Oncol Nurs. 2022, 9, 100128.
- Danilatou, V.; Nikolakakis, S.; Antonakaki, D.; Tzagkarakis, C.; Mavroidis, D.; Kostoulas, T.; Ioannidis, S. Outcome Prediction in Critically-Ill Patients with Venous Thromboembolism and/or Cancer Using Machine Learning Algorithms: External Validation and Comparison with Scoring Systems. J. Mol. Sci.2022, 23, 7132.
- Sanfilippo, K.M.; Wang, T.F.; Carrier, M.; Falanga, A.; Gage, B.F.; Khorana, A.A.; Maraveyas, A.; Soff, G.A.; Wells, P.S.; Zwicker, J.I. Standardization of risk prediction model reporting in cancer-associated thrombosis: Communication from the ISTH SSC subcommittee on hemostasis and malignancy. J Thromb Haemost. 2022, 20, 1920-1927
We hope that the current version of the manuscript appropriately addresses your suggestions and that now you will find our paper suitable for publication in Cancers.
Thank you for kindly considering our manuscript!
Looking forward to your decision!
Sincerely Yours,
Anca Drăgan
Corresponding author:
Anca Drăgan, Ph D, MD
1st Department of Cardiovascular Anaesthesiology and Intensive Care
“Prof. C.C. Iliescu” Emergency Institute for Cardiovascular Diseases
Sos. Fundeni, No 258, 022328, Bucharest Romania
Tel: +40760587855
E-mail: anca.dragan1978.14@gmail.com
Reviewer 2 Report
Comments and Suggestions for Authors
The authors describe the topic in an exhaustive and detailed way. Referring to recent guidelines, they underline the need to deepen risk assessment in outpatients with cancer.
The description of the application of different scores for VTE risk assessment is complete and well documented.
The application of algorithmic machine learning tools could be useful in future clinical practice, but not without optimization in larger databases.
The bibliography reported is updated and adequate for the descriptions.
The presentation of the different aspects is clear and easily understandable.
Author Response
Bucharest, January, 7th, 2023
To:
Reviewer 1
Cancers
Dear Professor,
We are submitting the revised version of our manuscript entitled “"Novel insights in venous thromboembolism risk assessment methods in ambulatory cancer patients: from the guidelines to clinical practice” to be considered for publication in Cancers as a Review. The initial version of our manuscript has been revised taking into consideration your comments.
Each point made by you is addressed below separately, explaining what we have changed and where the changes are to be found in the manuscript (the changes are highlighted in the revised manuscript).
We thank you for your useful suggestions and comments and your kind words about our manuscript. We greatly appreciate the time and efforts you have taken for the evaluation of our work!
Comments and Suggestions for Authors
The authors describe the topic in an exhaustive and detailed way. Referring to recent guidelines, they underline the need to deepen risk assessment in outpatients with cancer.
The description of the application of different scores for VTE risk assessment is complete and well documented.
The application of algorithmic machine learning tools could be useful in future clinical practice, but not without optimization in larger databases.
The bibliography reported is updated and adequate for the descriptions.
The presentation of the different aspects is clear and easily understandable.
Response
We thank very much you for your appreciation!
We have improved the Machine learning section by discussing the advantages and disadvantages of these methods and updated it with new references. The change was done at Reviewer 1suggestion.
We have added a figure that summarizes the facts exposed in the review at the suggestions of Reviewer 3.
We have also improved Table 1 by including the class and level of evidence for the guidelines recommendations. Additionally, we have added a new Table 2, which presents the relevant studies on the modalities and importance of VTE screening among ambulatory cancer patients, as suggested by Reviewer 3. We restructured Table 2 from the initial manuscript based on Reviewer 3 suggestions. The new Table 3 is more informative and includes data on the studied population, VTE incidence, main findings, and validation data.
The revised manuscript includes the changes that are highlighted.
We hope that the current version of the manuscript appropriately addresses your suggestions and that now you will find our paper suitable for publication in Cancers.
Thank you for kindly considering our manuscript!
Looking forward to your decision!
Sincerely Yours,
Anca Drăgan
Corresponding author:
Anca Drăgan, Ph D, MD
1st Department of Cardiovascular Anaesthesiology and Intensive Care
“Prof. C.C. Iliescu” Emergency Institute for Cardiovascular Diseases
Sos. Fundeni, No 258, 022328, Bucharest Romania
Tel: +40760587855
E-mail: anca.dragan1978.14@gmail.com
Reviewer 3 Report
Comments and Suggestions for Authors
In this study authors aimed to review the current relevant knowledge in VTE risk assessment in ambulatory cancer patients, starting from the guidelines recommendations and continuing with the specific risk assessment methods and machine learning models approaches.
With this purpose, authors made a narrative review.
Thanks for the opportunity to review this manuscript.
There are some limitations/comments that authors need to address:
The main limitation of this work lies in the narrative review, which, being non-systematic, may introduce bias in the inclusion or exclusion of certain studies. This limitation should be acknowledged at the end of the paper before the conclusion.
This study would benefit from English language revision and editing.
To enhance the impact of the work, I suggest creating a figure summarizing all the reviewed aspects. Visual abstracts or figures of this kind tend to capture the reader's attention and attract more views.
Similarly, I recommend summarizing the findings discussed throughout the text in tables (see comments below). I suggest that Table 1 include the level of evidence from the recommendation / suggestion of each guideline.
Section 3 need to include a table that summarize all studies with information about population include in each work.
Table 2 need to include more information (I. e. number of patients in each study, number (%) of VTE event, sex, ethnicity, age, metastasis, externally validated or not). This data provides to the readers information about population included in different studies.
Comments on the Quality of English LanguageThis study would benefit from English language revision and editing.
